# Construction of Antibody Phage Libraries and Their Application in Veterinary Immunovirology

**DOI:** 10.3390/antib9020021

**Published:** 2020-06-03

**Authors:** Shahbaz Bashir, Jan Paeshuyse

**Affiliations:** Department of Biosystems, Division of Animal and Human Health Engineering, Laboratory of Host Pathogen Interaction in Livestock, KU Leuven University, 3000 Leuven, Belgium; shahbaz.bashir@kuleuven.be

**Keywords:** phage display, immunovirology, monoclonal antibody, veterinary medicine

## Abstract

Antibody phage display (APD) technology has revolutionized the field of immunovirology with its application in viral disease diagnostics and antiviral therapy. This robust and versatile technology allows the expression of an antibody fused to a phage coat protein on the surface of a filamentous phage. The DNA sequence coding for the antibody is packaged within the phage, linking the phenotype to genotype. Antibody phage display inherits the ability to rapidly generate and modify or improve high-affinity monoclonal antibodies, rendering it indispensable in immunology. In the last two decades, phage-display-derived antibodies have been extensively used in human medicine as diagnostic and therapeutic modalities. Recently, they are also gaining significant ground in veterinary medicine. Even though these advancements are mainly biased towards economically important animals such as chicken, cattle, and pigs, they are laying the foundation of fulfilling the unmet needs of veterinary medicine as antibody-based biologics in viral diagnostics, therapeutics, and immunoprophylaxis. This review provides a brief overview of the construction of antibody phage libraries and their application in diagnosis, prevention, and control of infectious viral diseases in veterinary medicine in detail.

## 1. Introduction

Phage display has emerged as a very powerful, robust, and effective molecular technique to generate vast libraries containing millions or even billions of different peptides or proteins. It includes batch cloning of DNA encoding for millions of variants of certain ligands (e.g., peptides and proteins or fragments thereof) into the phage genome as part of one of the phage coat proteins (p Ⅲ, p Ⅵ, or p Ⅷ). Exposition of proteins or polypeptides on the surface of phage is achieved by fusion of the coding sequence of one of the coat proteins to the gene of interest, which helps in isolating the specific binding ligands by a series of recursive cycles of selection on antigen or ligand with each cycle comprising of binding, washing, elution, and amplification. Generally, proteins are displayed on phage particle tethered to either minor coat proteins like p Ⅲ or the major coat proteins like p Ⅷ, yet retaining its structure/function. Based on the direct linkage between phenotype and genotype, phage display technology (PDT) provides an expedient approach to study the genetics and functionality of the interacting proteins and peptides [1]. Following the initial demonstration of peptide display, antibodies were the first functional proteins to be successfully displayed on the phage surface. In a humoral immune response, antibodies serve as antigen-targeting effector molecules. Their unique tetrameric structure consists of two identical light and two identical heavy chains joined together by disulfide bridges and non-covalent interactions. Each chain is comprised of a series of discreet repeats, forming a compactly folded regions of proteins called “antibody domains” (Figure 1). Antibodies are modular protein defense systems possessing a paratope (variable domain), an antigen-binding site located at the upper tips of the “Y”-shaped structure. These paratopes identify the specific epitope displayed by the antigen and tag the microbe as well as the infected cell to be recognized by the immune system for neutralization. Therefore, these variable domains serve as paradigmatic proteinaceous scaffolds to be expressed on the phage surface for the isolation of novel binders against a myriad of antigens. Phage display of combinatorial antibody libraries is an efficient technique by which monoclonal antibodies (mAbs) of a desired specificity can be selected without the use of conventional hybridoma technology [2]. Their unique maturation process helps them to evolve to be highly specific to viral antigens, opening new horizons in viral disease diagnosis and therapeutics. In the last few decades, the prime focus of the developed recombinant antibodies was targeted towards human medicine, but now the shift in trends towards veterinary medicine seems promising. In spite of skewed research trends towards economically important livestock species such as cows, poultry, sheep, goats, and pigs, there is a vast field of veterinary medicine waiting for the application of mAbs developed through antibody phage display. Furthermore, efficient techniques have been established and optimized to design, build, and manipulate the vast antibody fragment-based libraries in order to derive the antibodies of desired characteristics and affinity. Here, we encompass the progress in this rapidly growing field and discuss its application in finding new diagnostics and therapeutic viral targets in veterinary medicine in association with other complementary technologies. 

## 2. Antibody Phage Display Library Construction and Biopanning

An antibody phage library is the collection of antibody variable domains displaying phages with library size 10^6^–10^11^, depending on the library type, fused to their coat proteins as a result of antibody-variable fragment (gene) cloning. Here, we shall briefly describe the construction of a phagemid-based antibody (scFv and Fab) phage display library from immunized or naïve sources. The basic methodology to build APD library from the V-gene repertoire (Figure 2) is almost the same, starting with the extraction of mRNA from B-cells isolated from blood, spleen, tonsils, and tumor tissue samples. Using oligo (dt) primers or random primers, cDNA is synthesized from mRNA through reverse transcription-polymerase chain reaction. Defined PCR primer sets, designed on the basis of species consensus sequence and anneal to the conserved region of V-gene families or constant domain, are used to amplify the heavy chain and light chain corresponding region (V_H_ and V_L_; V_H_, C_H_, V_L_ and C_L_, respectively) genes within a given immunoglobulin repertoire from cDNA pools, thus revealing the all antibody specificities in a particular individual [4]. The PCR-amplified V_H_ and V_L_ and V_H_-C_H_ and V_L_-C_L_ gene fragments are ligated in a suitable phagemid for single-chain variable fragment (scFv) and Fab library generation, respectively. In the case of synthetic antibody phage libraries, the initial few steps are not needed, and library diversity is increased through the precise introduction of degenerate DNA into CDR encoding regions, thus rivaling or exceeding than that of natural immune repertoire [5]. 

The engineered phagemids are transformed into suitable bacteria (e.g., XL1-Blue, TG1, and ER2537), providing a suitable environment for recombination of antibody fragments [6]. For rescuing the recombinant phagemid harboring the gene of inserts like Fab or scFv, the transformed bacteria are infected with helper phages like VCSM13 or M13KO7. These phages belong to the M13 class of bacteriophages and are well-adapted for the exposition of antibody-variable scaffolds (Figure 3). This results in a library of phages, where each phage is expressing a unique antibody fragment on its surface as a phenotype while possessing the vector with specific nucleotide sequences within as respective genotypes [7,8].

Now, this antibody phage library is ready to pan against the antigen of choice. By using this methodology, antibody phage libraries of different antibody formats have been successfully constructed, such as Fab (antigen-binding fragment), Fv (variable fragment), scFv (single-chain variable fragment), and its modifications [9,10] diabodies, and other oligomers [11,12]. The science of selection of high-affinity clones is as important as the generation of large, diverse, and more rational antibody libraries. Antigen-specific clones, phages displaying antibody fragments, are screened and enriched from antibody libraries through an iterative process based on interaction affinity, avidity, and binding kinetics between the expressed antibody fragment and the antigen of interest called biopanning. It includes recursive cycles of pooling of antibody displaying phages, incubation with antigen, removal of unbound phages (washing), elution of bound phages, re-amplification in F^+^
*E. coli*, and re-selection (Figure 4) [13]. It represents as a high throughput antibody library screening strategy allowing screening of a library with up to 10^11^ variants and also isolation and characterization of high-affinity binders with a frequency as low as 1 in 10^6^ by mimicking the same process by which immune system selects antibodies against a specific antigen through affinity maturation. A strong combination of interactive affinity selection and biological amplification controls the overall strength of the biopanning procedure. There are various types of biopanning strategies in practice, but selecting only one is very much dependent on the nature, functionality, and native environment of the antigen. The choice of screening method can directly influence the antibody phage output number and diversity among selected clones. Therefore, it should be adapted to fit the respective class of antigen [14]. Biopanning strategies against pure or soluble antigens are well established and have been performed against pure or non-membranous protein antigens by immobilizing it to a solid support through adsorption, such as immunotubes [15,16], microtiter plates, and BIAcore sensor chips [17,18,19,20]. On the other hand, membrane-associated proteins, possessing hydrophobic domains, account for 20%–30% of all protein in living entities, thus presenting themselves as desirable targets for APD in order to obtain novel mAbs for research, diagnostic, and therapeutic purposes [21]. Direct panning on whole cells, displaying the complex target antigen on their surface, alongside existing ligand guided “pathfinder” biopanning strategy, has proven to be a reasonable modality in selecting high-affinity antibody clones against membrane proteins [22,23]. Alternatively, detergent micelles and nanodiscs have evolved as an interesting substitute for membrane protein expression for the isolation of novel antibody binders [24,25,26,27]. In recent times, virus-like particles (VLPs) have emerged as an interesting platform allowing multivalent display of membrane proteins and are yet to be explored for the screening of novel binders against membrane target proteins [28]. In spite of this progress in membrane protein expression for the pursuit of new antiviral targets, mechanistic linkages between a protein’s conformational transitions and its function pose a significant challenge.

### Quality Assessment of an Antibody Library

The performance of an antibody library is directly linked to its quality. The salient features governing the library quality control include (a) the number of clones with phagemids harboring the insert, i.e., scFv or Fab, (b) the number of clones expressing phages carrying inserts, and (c) the number of clones capable of soluble expression of inserts. The key parameter of an antibody library’s quality is its complexity, also called diversity. It is an estimate of distinct elements in that collection. It directly reflects the probability of screening an antibody clone from the library against a given antigen, of sufficiently higher affinity. Theoretically, the probability of finding a functional antibody against a given antigen is higher in large complex or more diverse libraries than in a less complex or diverse library. It is approximated by the transformation efficiency of the bacteria used to amplify the library. Despite the simplicity and significance of this concept, the accurate and reliable quantification of this prime feature was not possible and is still lacking [29,30]. PCR screening of individual clones to determine the presence of insert, dot plot analysis to detect both antibody fragment displaying phages, and soluble antibody fragment synthesized by the screened clones, and further characterization of the few hundred positive clones through DNA fingerprinting and sequencing are still considered standard practices [31,32]. Recently, a reseach group used a PCR-free next-generation sequencing (NGS) approach coupled with new bioinformatic tools for the reliable and accurate estimation of the diversity of a library [33]. However, further optimization and advancement in NGS is required to fully exploit the potential of this incredible technology in tackling the antibody library diversity quality assessment.

## 3. Application of APD in Veterinary Immunovirology

Immunovirology, also called viral immunology, encompasses the evolution of the immune system as a result of its interaction with viruses. In 1996, Zinkernagel published an article titled “Immunology taught by viruses”, explaining this incredible interaction between viruses and the immune system [34]. Despite the fact that previously, human medicine was the focal point of recombinant antibodies development through antibody phage display, recent advancements in veterinary medicine are promising. In the field of veterinary medicine, the impact of viral infections on animal welfare and health cannot be neglected. Major strides have been taken in veterinary diagnostics. Here, we review the application of phage-display-derived antibodies in veterinary viral diagnosis, therapeutics, and immunoprophylaxis. These antibodies are also described in Table 1 with respect to format, species of origin, and target antigen.

### 3.1. Veterinary Diagnosis

By using the inherent selection potential of antibody phage display, murine scFv capable of recognizing the avian influenza virus (AIV) was isolated and evaluated through ELISA. It showed higher sensitivity and specificity as compared to the previously established protocols [35]. An antibody repertoire of llamas immunized with purified nucleoprotein (AIV-NP) of AIV yielded a novel nanobody (VHH). This VHH led to the development of a lateral flow assay for the rapid detection of AIV [36]. Mining of the rabbit antibody repertoire through antibody phage display yielded a novel scFv capable of targeting a conserved motif in the C-terminal of the F2 protein of Newcastle Disease Virus (NDV). This scFv has exceptional diagnostic applications, ensuring rapid pathotyping of NDV isolates [37]. Chicken antibody repertoire has also been explored with phage display for the screening of scFv against phosphoprotein (NDV-P) of NDV. This protein is involved in viral RNA replication and transcription, thus presents itself as a potential diagnostic target in control of NDV. An isolated scFv showed higher specificity for NDV-P by ELISA and Western blot [38]. Another important virus causing considerable economic losses to the poultry industry is the infectious bursal disease virus (IBDV). Due to a variety of IBDV strains, it was always difficult to distinguish between very virulent IBDV (vvIBDV), classical, variant, and vaccine strains. Sapats and colleagues (2006) isolated a single-chain variable fragment, which efficiently recognizes a highly conformational epitope of VP2 protein of vvIBDV. The efficacy of this mAb in distinguishing vvIBDV from other strains makes it ideal for diagnostic tests like ELISA [39,40]. Besides chicken, duck has also become a significant part of the worldwide poultry industry. One of the biggest challenges faced by this industry is duck viral hepatitis (DVH) caused by duck hepatitis A virus (DHAV). It is characterized into three serotypes, namely, DHAV-1, DHAV-2, and DHAV-3. Out of these serotypes, DHAV-1 is the most virulent, causing significant widespread losses to this growing industry. Through phage display, a novel nanobody has been isolated after the screening of the camelids antibody repertoire, targeting the VP1 protein of DHAV-1. This nanobody has wide applications in the development of immunoassays for the DHAV-1 diagnosis, such as ELISA [41]. 

In the dairy and beef industry, foot and mouth disease (FMD) is considered the most devastating viral disease caused by foot and mouth disease virus (FMDV), inflicting huge economic losses. There are seven serotypes of FMDV, including O, A, C, Asia 1, SAT 1, SAT 2, and SAT 3. New control programs have adopted a combined strategy of exclusion, slaughter, and vaccination. However, the success rate depends upon accurate and reliable differentiation between vaccinated and infected animals. Previously through phage display, murine scFv was isolated against FMDV type O, but it lacked the diagnostics potential in differentiating infected from vaccinated animals [42]. To circumvent this bottleneck, recombinant chicken scFv has been isolated against 3ABC protein of FMDV through APD, and its application in ELISA differentiates infected from vaccinated animals [43,44]. Additionally, murine scFv anti-VP2 of FMDV has demonstrated an efficient diagnostic potential for several FMDV serotypes [45]. In the case of the bovine immunodeficiency virus (BIV), murine recombinant antibody in scFv format was generated and showed potential as a detection tool in competitive inhibition ELISA for serological detection of BIV infection in bovines [46].

In pigs, the initial pursuit for new diagnostic tools using recombinant Abs (rAbs) has focused on porcine circovirus type II (PVC2) and classical swine fever. A nanobody (VHH) was obtained from camel immunized with a commercial PVC2 vaccine through APD. The fusion of this VHH with alkaline phosphatase has proved to be an added value in PVC2 diagnosis without any cross-reactivity to PVC1 and porcine reproductive and sensitivity virus (PRRSV) [47]. The nanobody has already shown tremendous potential as a promising experimental tool in imaging of porcine epidemic diarrhea virus (PEDV) [48]. In addition to this, a highly potent porcinized murine antibody yielded favorable results in diverse diagnostic assays for classical swine fever, while retaining in vitro neutralizing ability [49]. Recently, the swine influenza virus (SIV) and PEDV have gained a lot of attention for generating novel binders of diagnostic value through phage display. SIV is a significant pathogen posing serious challenges to swine production. Due to the limitations associated with previous diagnostics tools of SIV infection such as subjectivity, high cost, and lack of suitability on a large scale, VHH was developed through phage display against the conserved nucleoprotein of SIV offering applications in a wide range of immunoassays for SIV diagnosis [50]. Similarly, in two separate studies, VHHs were isolated against different target antigens of PEDV, enabling its rapid and sensitive clinical detection. These antibodies are of great added value in the success of the ongoing PEDV control program. PEDV causes huge production losses to the pork industry through a fatal diarrheal disease infecting mainly neonatal and post-weaning piglets [51,52].

In dogs, a canine antigen-experienced antibody repertoire has been explored for isolation of scFv with great diagnostic potential against canine parvovirus disease [53]. This certainly paves the way for future exploration of canine antibody repertoire for certain applications in canine medicine.

In horses, the western equine encephalitis virus (WEEV) and the Venezuelan equine encephalitis virus (VEEV) are zoonotic viruses, constantly posing a threat of widespread equine and human epidemics. In early infection of VEEV and WEEV, viral detection is considered as an indicator for the implementation of appropriate prophylaxis such as vaccination to decrease the severity and rapid transmission of the disease. In the pursuit of the ideal detection tools, excavation of murine antibody repertoire has resulted in the isolation of novel scFv, having extensive applications in immunoassays, immunohistochemistry, and radio-immunodiagnostics [54,55]. 

In sheep, the Maedi-visna virus (MVV), an enveloped non-oncogenic RNA virus, causes a complex syndrome characterized by encephalomyelitis, arthritis, and pneumonitis. Antibody phage display was applied to the synthetic human scFv Griffin.1 library for the isolation of a high-affinity scFv antibody against capsid protein and conserved principal immunodominant domain of transmembrane protein gp46 protein in two separate studies. These scFvs have been proven to be efficient research and diagnostic tools in MVV control programs [21,56]. In addition to this, an scFv of in vitro veterinary diagnostic importance has been isolated against the blue-tongue virus (BTV) from a large chicken semi-synthetic “Nkuku” antibody phage library [57].

### 3.2. Veterinary Therapeutics

The role of rAbs in veterinary therapeutics is very limited, indicating the need for its expansion by finding new clones capable of targeting viral disease. Mostly in the poultry sector, the therapeutic application of rAbs is considered economically impractical. A prevention strategy against influenza viral infection has been tested by using recombinant adenovirus expressing neutralizing VHH in mice models. An initial study has shown promising results, with a 90%–100% survival rate in lethally challenged mice [58]. Similarly, a Fab fragment isolated from the chicken antibody repertoire has delineated encouraging results by targeting HA0 hemagglutinin epitopes of highly infectious H5N1 [59]. Recently, anti-DHAV-1 scFv has been obtained from a murine antibody repertoire, which harbors the ability to neutralize DHAV-1 by targeting its VP3 protein. The scFv has shown promising therapeutic effects against duck viral hepatitis, a highly infectious disease of ducks with a mortality rate of up to 100% [60].

Bovine viral diarrhea virus (BVDV) is a significant cattle pathogen, inflicting huge economic losses to the livestock industry worldwide. It causes a complex syndrome involving multiple body organs. The sign and symptoms include diarrhea, pneumo-gastroenteritis, abortion, stillbirth, and births of persistently infected calves. To combat this infectious virus, it is imperative to establish novel antivirals to mitigate the devastating economic impinge of this disease in the cattle industry. In this scenario, BVDB NS5B-specific nanobody was obtained through phage display, which is capable of significantly suppressing in vitro BVDV replication. It has demonstrated a promising potential to be an effective anti-BVDV agent [61]. Furthermore, in a separate study, a specific nanobody was developed against the E2 protein of BDVD, a protein involved in viral pathogenesis and capable of producing neutralizing antibodies. The efficient binding alongside in vitro neutralization ability makes this anti-E2 nanobody a promising therapeutic candidate in controlling BVDV infection worldwide [62].

In pig farming, transmissible gastroenteritis virus (TGEV) is considered a highly dangerous infectious agent that causes watery diarrhea, vomiting, and ultimately dehydration in pigs. Notably, in piglets, a 100% mortality rate has been observed post-TGEV infection. Recently, a porcine scFv antibody has been identified and characterized, having in vitro TGEV neutralizing ability. The therapeutic potential of this novel binder is promising in the control of porcine viral gastroenteritis [63]. Similarly, high-affinity nanobodies targeting Nsp4 and Nsp9 protein of PRRSV with therapeutic potential have been isolated and characterized [64,65].

The exploitation of a naïve llama antibody repertoire resulted in the screening of rabies virus glycoprotein specific VHH, which later led to the generation of homogenous pentavalent multimers called combodies, possessing increased virus-neutralizing capability, following the fusion with coiled–coil peptides from human cartilage oligomeric matrix protein [66]. In dogs, the therapeutic application of rAbs is mainly restricted to cancer and inflammatory disturbances [72]. 

In horses, equine herpesvirus-1 (EHV-1) negatively impinges their welfare and health by causing abortion and respiratory and neurological diseases. Molinkova and colleagues scanned the synthetic human Griffin.1 antibody library for the selection of a highly potent scFv antibody against EHV-1. In vitro EHV-1 neutralization ability of the screened scFv clone makes it an ideal therapeutic candidate [67].

In human medicine, the therapeutic antibodies account for a market of about USD 75 billion [73]. In contrast, there exists a great economic disparity when it comes to the therapeutic application of rAbs in veterinary medicine. This demands a continuous search for alternative ways to control viral diseases using rAbs in veterinary medicine by focusing on all animal species.

### 3.3. Veterinary Immunoprophylaxis

In terms of FMDV prophylactic measure, initial clinical trials of the in vivo application of nanobody-based drugs have shown promising results in guinea pigs, providing them with partial protection against FMDV type O [68]. Besides cattle, FMDV infects a wide range of cloven-hoofed animals, including pigs, goats, and sheep. Passive immunization of pigs with a mixture of two novel nanobodies provided them with complete rapid protection from a FMDV challenge, thus improving the FMD immunoprophylactic program [69]. Until now, the high level of serotype specificity of these nanobodies renders them exclusive to pigs, thus limiting their application to other species. Furthermore, passive immunization of piglets with anti-PEDV scFv screened from a porcine antibody repertoire has demonstrated encouraging results by providing them complete protection form PEDV infection [70]. To date, the implementation of the immunoprophylactic potential of the phage display selected antibodies is limited to pigs; however, the spread of this approach is expected in other species as well in the near future.

## 4. Conclusions and Future Perspectives

The inherent potential of combinatorial libraries gives researchers unprecedented control over the expression of the acquired immune system. To date, the prime focus of rAbs has been in veterinary diagnostic applications, ignoring the utilization of their immense therapeutic and immunoprophylactic potential. The diagnostic potential of these monoclonal antibodies is well appreciated, but their therapeutic application against infectious diseases is still a challenge. Although the evolution of antibody engineering has led to highly tailored antibodies with improved pharmacokinetic properties, yet experimental therapeutics remains challenging. For example, the advantages associated with scFv account are a curse in disguise rather than a blessing and render it undesirable for therapeutic application. Small size, low in vivo half-life, rapid clearance, and low stability are the hurdles in the therapeutic avenue. Nevertheless, PEGylation and conjugation of scFv with certain drugs and toxins have circumvented some of the issues for their therapeutic application, but a lot still has to be done. Furthermore, only a few rAb forms have been elucidated in veterinary medicine such as scFv and VHH, while the diagnostic and therapeutic potential of other forms is yet to be explored. In vaccinology, vectored-based scFv delivery for in vivo expression has come out as an effective strategy to sustain the level of antibody fragments in the body. Furthermore, broadly neutralizing antibodies against viral diseases alongside in vivo scFv delivery systems have paved a new avenue of functional reagents, making them more relevant in therapeutics [74]. Additionally, advancements in the targeted delivery of rAbs to various novel compartments of the animal body such as the gut, brain, or to the intracellular cytosolic compartment could certainly have a huge impact on the efficacy of antibody therapeutics [75]. In the veterinary field, these are just a few examples of implementation of APD, while the vast diagnostic and therapeutic potential of diverse antibody repertoire amongst different species is yet to be explored in relation to the animal diseases. Furthermore, the vast scope of APD technology also encompasses the other areas of immunovirology, such as understanding the pathogenesis of viral infections and their contributing factors. With every passing day, the implementation of this incredible technique is opening new dimensions in immunovirology in terms of pathogenesis, diagnosis, and therapeutics.

## Figures and Tables

**Figure 1 antibodies-09-00021-f001:**
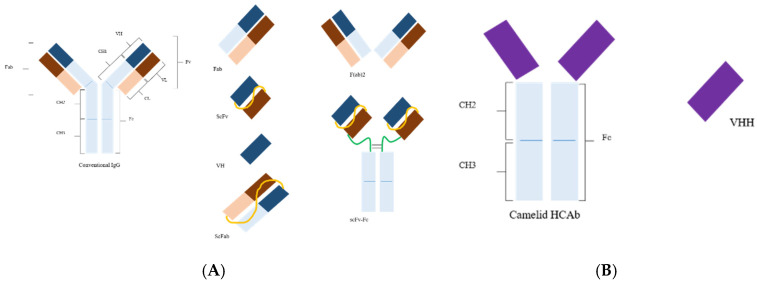
(**A**) Structure of a conventional antibody and its derivatives. Abbreviations: Fv, fragment variable; Fc, fragment crystallizable; Fab, fragment antigen binding; scFv, single-chain variable fragment with linker (yellow); scFab, single-chain Fab; VH, variable domain heavy chain; F(ab)2, two Fab fragments joined together; scFv–Fc, scFv linked to Fc (inter- and intra-disulfide linkages are not shown here in the all formats). (**B**) Camelid heavy chain antibody (HCAb); VHH, nanobody (modified from [3]).

**Figure 2 antibodies-09-00021-f002:**
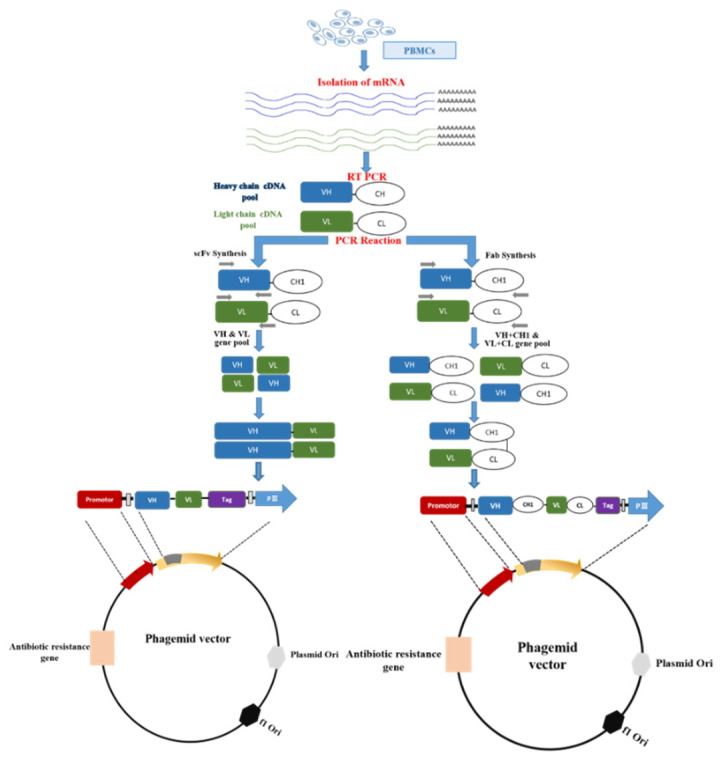
Synthesis and cloning of scFv and Fab fragments in a phagemid vector. It starts with the preparation of peripheral blood mononuclear cells (PBMCs) using density gradient from whole blood. Following RNA extraction from PBMCs, cDNA is synthesized through reverse transcription. From cDNA, the variable region of heavy (VH) and light chain (VL) is amplified through PCR using specific primers. scFv and Fab are constructed through splicing by overlap extension PCR(SOE-PCR). The amplified scFv and Fab are ligated into the phagemid vector following restriction digestion of both.

**Figure 3 antibodies-09-00021-f003:**
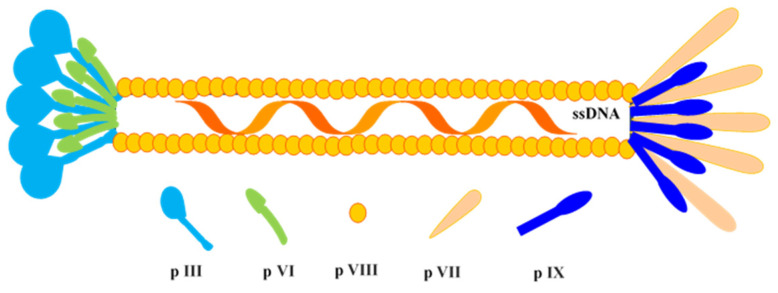
Schematic diagram of M13 phage. It carries ~6.4-kb sized circular single-stranded DNA (ssDNA), which encodes for 10 proteins, i.e., p I, p II, p III, p IV, p V, p VI, p VII, p VIII, p IX, and p X.

**Figure 4 antibodies-09-00021-f004:**
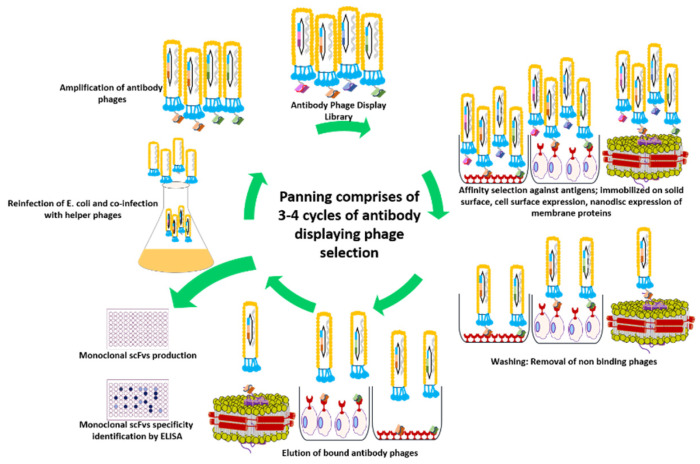
Panning of an antibody phage display library for the generation of high-affinity antibodies. The antigen is presented in its native form either immobilized on a plastic surface or expressed on the cell surface. For membrane protein targets, nanodisc provides a membrane-like environment. Antibody library displaying phages are incubated with antigen, and non-bound or weakly bound phages are removed by washing. The tightly bound phages are recovered by elution via trypsin or pH shift for the re-infection of *E. coli* cells. Following the coinfection with helper phages, new phage particles are produced for usage in subsequent rounds of panning. This cycle continues for 3–4 rounds, leading to enrichment of binders.

**Table 1 antibodies-09-00021-t001:** Summary of the phage display derived antibodies with their potential use in veterinary medicine.

Application of Antibody	Antibody Format	Source Species	Target	Reference
Diagnosis	scFv	Mouse	Recombinant NP protein of avian influenza virus in poultry	[35]
	VHH	Camelid	NP protein of AIV in poultry	[36]
	scFv	Rabbit	F2 protein of Newcastle disease virus in poultry	[37]
	scFv	Chicken	Phosphoprotein of Newcastle disease virus in poultry	[38]
	scFv	Chicken	Whole infectious bursal disease virus in poultry	[39,40]
	VHH	Camelid	VP1 protein of duck hepatitis A virus-1 in ducks	[41]
	scFv	Mouse	Foot and mouth disease virus type O in cows	[42]
	scFv	Chicken	3ABC protein of foot and mouth disease virus in cows	[43,44]
	scFv	Mouse	VP2 of protein of foot and mouth disease virus in cows	[45]
	scFv	Mouse	Capsid protein of bovine immunodeficiency virus in cows	[46]
	VHH	Camelid	Capsid protein of porcine circovirus type in pigs	[47]
	VHH	Camelid	Membrane proteins of porcine epidemic diarrhea virus in pigs	[48]
	scFv	Mouse	E2 protein of classical swine fever virus in pigs	[49]
	VHH	Camelid	P protein of, swine influenza virus in pigs	[50]
	VHH	Camelid	N protein and spike protein of porcine epidemic diarrhea virus	[51,52]
	scFv	Dog	Capsid protein of canine parvovirus in dogs	[53]
	scFv	Mouse	Whole Western equine encephalitis virus	[54]
	scFv	Mouse	Whole Venezuelan equine encephalitis virus	[55]
	scFv	Human *	transmembrane envelope glycoprotein gp46 of Maedi-visna virus in sheep	[21]
	scFv	Human *	P25 of Maedi-visna virus in sheep	[56]
	scFv	Chicken	Whole blue-tongue virus in sheep	[57]
Therapy	VHH	Camelid	HA antigen of H5N1 in poultry	[58]
	Fab	Chicken	HA0 hemagglutinin of H5N1 in poultry	[59]
	scFv	Mouse	VP3 protein duck hepatitis A virus-1 in ducks	[60]
	VHH	Camelid	NS5B protein of bovine viral diarrhea virus in cows	[61]
	VHH	Camelid	E2 protein bovine viral diarrhea virus in cows	[62]
	scFv	Pig	Whole transmissible gastroenteritis virus in pigs	[63]
	VHH	Camelid	non-structural protein 4 of porcine reproductive and respiratory syndrome virus in pigs	[64]
	VHH	Camelid	non-structural protein 9 of porcine reproductive and respiratory syndrome virus in pigs	[65]
	VHH	Camelid	Whole rabies virus in dogs	[66]
	scFv	Human *	gD protein of equid herpesvirus-1 in horses	[67]
Immunoprophylaxis	VHH	Camelid	Whole food and mouth virus type O in cows	[68]
	VHH	Camelid	Whole food and mouth virus in cows pigs	[69]
	scFv	Pig	Spike protein of porcine epidemic diarrhea virus in pigs	[70]

* Synthetic human Griffin.1 [71].

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
