# Peer review of "Construction of Antibody Phage Libraries and Their Application in Veterinary Immunovirology"

_2073-4468, 2020, doi:10.3390/antib9020021_

Round 1
Reviewer 1 Report
This is a nice, comprehensive review.There are some grammatical errors and typos which need to be corrected - for eg: "Food and mouth disease" will be "Foot and mouth disease"
Author Response
Dear madam, sir,
We thank the reviewer for her/his time and effort in reviewing our manuscript. We made a thorough revision and annotated our changes in the included word file using track changes.
Best regards,
Jan Paeshuyse

Reviewer 2 Report
The review consists of two arms: 1) APD and 2) its application in veterinary immunovirology. It aggregates two “half-done” arms.
The first arm is the weak part of the article and should be significantly diminished. The second arm can be improved and form the thrust of the paper.
Arm 1:
There are far better informative and authoritative comprehensive review articles on APD in the public domain than what is presented here. Therefore, I do not see the added benefit of this arm especially when it arbitrary goes into some aspects of APD technology leaving other aspects uncovered. Therefore, for this part, the authors should give a very brief, overall and conceptual view of the subject matter staying away from technical or biological aspects as presented here as much as they can, referring the readers instead to excellent reviews already in the public domain on the subject matter. In line with this, figures should also be simple, again capturing concepts without getting into unnecessary details.
Arm 2:
Arm 2 is where the article can be of benefit to readers and where the authors should put their focus on, making it the main thrust of the paper. For this part, the authors may need to expand on what they have already covered and further cover any related studies that they might have missed covering here.
Author Response
Dear madam, sir,
We thank the reviewer for her/his time and effort in reviewing our manuscript. Per suggestion of the reviewer we now reduced the first part of the review and elaborated more on the immunovirology part of the section. We made a thorough revision and annotated our changes in the included word file using track changes.
Best regards,
Jan Paeshuyse

Reviewer 3 Report
The manuscript entitled ‘ Construction of antibody phage libraries and its application in veterinary immunovirology’ provides an overview of the phage-displayed antibody library and its application in veterinary diagnosis and therapeutics.
Generally, this is an interesting review, especially regarding veterinary Immunovirology. There are some grammatical errors. For example, line 288 ’ humans medicine’; line 292, ‘the application of antibody phage display derived antibodies’. The authors are supposed to be very careful and revise the whole manuscript again. Besides, I will suggest, it would be better to minimize the part of library construction and selection which have been summarized many times over the past two decades and try to introduce something new. It is a good idea to give more on the part of veterinary immunovirology.
- Line 17, I will suggest saying ‘ the DNA sequence coding for the antibody fragment’.
- Figure 1. What is the line between fab HC and LC? The authors should indicate these in the legend.
- Line 270, I will suggest saying ‘theoretically’ rather than ‘ directly’.
- Line 288 and 289, antibody drugs are derived from many ways including display libraries. And the whole sentence is confusing.
- Line 305, say ‘scFv’ if you mentioned this before.
- In the section of veterinary diagnosis, the author gave some examples like pigs and dogs. What about other animals? And the author gave an example of the dairy industry. What about other industries? And there are no references to support ‘dogs’.
Author Response
Dear madam, sir,
We thank the reviewer for her/his time and effort in reviewing our manuscript. Per suggestion of the reviewer we now reduced the first part of the review and elaborated more on the immunovirology part of the section. We made a thorough revision and annotated our changes in the included word file using track changes.
To your specific comments we answer as follows:
"There are some grammatical errors. For example, line 288 ’ humans medicine’; line 292, ‘the application of antibody phage display derived antibodies’. The authors are supposed to be very careful and revise the whole manuscript again."
Response: We now made a thorough proof-reading of the manuscript in addition to amending all spelling errors pointed out by the reviewer.
"Besides, I will suggest, it would be better to minimize the part of library construction and selection which have been summarized many times over the past two decades and try to introduce something new. It is a good idea to give more on the part of veterinary immunovirology."
Response: Per suggestion of the reviewer we now reduced the first part of the review and elaborated more on the immunovirology part.
- Line 17, I will suggest saying ‘ the DNA sequence coding for the antibody fragment’.
Response: We now added this suggestion.
- Figure 1. What is the line between fab HC and LC? The authors should indicate these in the legend.
Response: We replied to your question and made the necessary changes as per suggestion of the reviewer. Please, see annotated manuscript.
- Line 270, I will suggest saying ‘theoretically’ rather than ‘ directly’.
Response: We now added theoretically instead of directly.
- Line 288 and 289, antibody drugs are derived from many ways including display libraries. And the whole sentence is confusing.
Response: we now rephrased this sentence.
- Line 305, say ‘scFv’ if you mentioned this before.
Response: We now mention scFv
- In the section of veterinary diagnosis, the author gave some examples like pigs and dogs. What about other animals? And the author gave an example of the dairy industry. What about other industries? And there are no references to support ‘dogs’.
Response: We now expanded upon this section including more species and added the references that were missing.
Best regards,
Jan Paeshuyse

Round 2
Reviewer 2 Report
The manuscript is vastly improved.
Abstract is not balanced:
While the body of the manuscript is well balanced now [between 1) the APD part and 2) APD application], the Abstract does not reflect that (the Abstract is by and large on the APD technology section). The authors should make the Abstract ~half and half. They should reduce the APD technology part and add for the Application part.
Include this study if it fits:
Nielsen K et al. Prototype single step lateral flow technology for detection of avian influenza virus and chicken antibody to avian influenza virus. J Immunoassay Immunochem. 2007;28(4):307-18.
The paper will be publication worthy once the above and below points are adequately addressed.
Other minor points:
- “Construction of antibody phage libraries and its application in veterinary immunovirology”. Replace “its” with “their”
- P2, L42: Replace “similar” with “identical” in “ two similar light”
- P2, L42: Replace “similar” with “identical” in “two similar heavy chains”
- P2, L45: Delete “a” in “Antibodies are a modular protein defence system”
- Figure 1 legend: 1) Fc, is not fragment constant. It is fragment crystallizable; 2) VHH is not derived from the typical IgG as you imply. 3) Divide you figure into “a” and b”. For “a”, keep your conventional IgG, scFv, ScFv-Fc, Fab, F(ab)2 and scFab. Remove VH/VHH. For “b’, draw a typical camelid heavy chain IgG and from it show its VHH derivative. 4) There are numerous disulfide linkages involved in these antibody fragments but you have chosen to show a few. I suggest you do not show any disulphide linkage and instead say: ”Inter- and intra-disulphide linkages are not shown”; 5) “VHH, variable domain heavy chain of camels” this means it could also be derived from conventional camelid IgGs which is not true. Change to: “VHH, variable domain of camelid heavy chain IgGs”
- p3, title: Add “display’ after “phage” in “Antibody phage library Construction and Biopanning”.
- Antibody phage library Construction and Biopanning Section: the library construction description is very specific here: it is on ScFv and Fab, not on VH, VL or VHH libraries. It also describes immune and naïve libraries not synthetic libraries which does not involve the first few steps you are describing. It also described phage display libraries that use phagemid and not phage vector. These need to be acknowledged in the section to avoided misleading readers into understanding that what you are describing is general to all libraries.
- Figure 4: 1) In the legend you have “PBMC” but in the figure itself “B-Lymphocytes”. Be consistent, stick to one. 2) There is no background information on V/D/J and V/J and you do not need to include it. Remove any reference to V/D/J and V/J and stick only to VH and VL. 3) In the right, Fab arm of your figure, remove “1” from “VL+CL1 gene pool”. 4) Replace “as” with “are” in “The amplified scFv and Fab as ligated”.
- Figure 2 legend. 1) Change “̴ 6.4 kb long” to “̴ 6.4 kb size”. 2) Include space in “DNA(ssDNA)”.
- Figure 5 legend: 1) Add “of” after “generation” in “Panning of an antibody phage display library for the generation”. 2) Replace “The antigen is presented ... or expressed in cell surface” with “The antigen is presented ... or expressed on cells.” 3) Replace “Antibody library … through the process called washing.” With ““Antibody library … by washing.” 4) Replace “The tightly bound…by process called elution…” with “The tightly bound…are recovered by elution …”. 5) Replace “… 3-4 rounds leading to production of high affinity monoclonal antibody.” With “… 3-4 rounds leading to enrichment of binders.”
- Replace “The majors strides has been taken” with “The major strides have been taken”.
- Replace “Here, we mention the application of antibody” with “Here, we review the application of antibody”.
- Break “Here, we mention the application of antibody phage display derived antibodies in veterinary viral diagnosis, and therapeutics and immunoprophylaxis and these antibodies are also summarized in table 1 explaining the format, species of origin and target antigen.” in to sentences and modify: “Here, we review the application of antibody phage display derived antibodies in veterinary viral diagnosis, therapeutics and immunoprophylaxis. These antibodies are also described in table 1 with respect to format, species of origin and target antigen.”
- Replace “murine single scFv capable of recognizing” with “ a murine scFv capable of recognizing”.
- Replace “Mining…has already yielded a novel scFv” with “Mining…yielded a novel scFv”
- Add “of” after “terminal” in “motif present at the C-terminal the F2 protein”
- Modify “Besides murine and rabbit antibodies, chicken antibody repertoire itself has been explored with phage display for the screening of scFv against phosphoprotein (P) of NDV.” To “Chicken antibody repertoire has also been explored with phage display for the screening of scFv against phosphoprotein (NDV-P) of NDV.”
- Replace “Isolated scFv has shown higher specificity for NDV-P by ELISA and western blot [36].” With “An isolated scFv showed higher specificity for NDV-P by ELISA and western blot [36].”
- This sentence is vague: “difficult to distinguish between very virulent IBDV (vvIBDV), classical, variant and vaccine strain…” Are all these four are different strains? If so, replace “strain” with “strains”.
- Break and modify “Although, new control programs have adopted a combined strategy of exclusion, slaughter and vaccination; however success rate depends upon accurate and reliable differentiation of vaccinated and infected animals.” To “New control programs have adopted a combined strategy of exclusion, slaughter and vaccination. However, success rate depends on accurate and reliable differentiation between vaccinated and infected animals.”
- Replace “Previously through phage display, mice scFv was isolated” with “Previously through phage display, a murine scFv was isolated”
- Did you define FMDV before its mention in “against FMDV type O”?
- Replace “In case of Bovine” with “In the case of Bovine”
- Replace “the initial pursuit of new diagnostic“ with “the initial pursuit for new diagnostic“.
- Replace “Abs (rAbs) have been focused” with “Abs (rAbs) has focused”
- “from camel immunized with commercial”. Is it camel or llama or alpaca? Also, in your table you only have camel as the source of all your VHHs. Is that true?
- “A single domain antibody (sdAb) was obtained from camel”. Use VHH here and in other places as they are all VHHs not other types of sdAbs. Also in your figure you specifically describe VHH only.
- Replace “sdAb has already shown tremendous potential” with ““the VHH has already shown tremendous potential”. I am assuming here that you are referring to the VHH isolated above and not to sdAbs in general.
- “…as a promising experimental tool in imaging and porcine epidemic diarrhea virus (PEDV)”. Is “and” supposed to be “of”?
- “highly potent procinized murine antibody”. “procinized” or “porcinized”?
- Replace “swine influenza virus (SIV) and PEDV has” with “swine influenza virus (SIV) and PEDV have”.
- Replace “In Sheep” with “In sheep”.
- Replace “These scFvs have proved” with “These scFvs have been proven”.
- “…tool in control program of MVV.” “…tool in MVV control programs”?
- Replace “veterinary therapeutics in very limited” with “veterinary therapeutics is very limited”.
- Replace “It has shown promising therapeutic effects against” with “The ScFv has shown promising therapeutic effects against”
- “…specific nanobody was”. “VHH” instead of “nanobody”? See also other places for possible replacement of nanobody with VHH. If you are going to use “nanobody”, make sure you clarify it to be VHH in brackets in its first mention.
- Improve the structure of the sentence “The efficient binding alongside in vitro neutralization ability this anti-E2 nanobody make it a promising therapeutic candidate in controlling BVDV infection worldwide.”
- Replace “respiratory and neurological disease” with “respiratory and neurological diseases”. There could be other places where you may need to replace “disease” with “diseases”
- Replace “ways to control viral disease” with “ways to control viral diseases”
- Table 1. Be consistent: 1) Chicken and rabbit are singular, mice is plural. Replace “mice” with “mouse”. 2) If it is dog, then replace “canine” with “dog”; 3) since you use “diagnosis”, then replace “Therapeutics” with “Therapy”. B. 1) They cannot be all camel. At least some must be llama. Address all “camels”. 2) Replace “Synthetic human Griffin. 1“ with “human“, then in the footnote say: ”Synthetic human Griffin. 1” and include reference. 3) To be consistent with your antibody domain abbreviation style, e.g., VL, CL, CH1, use VHH instead of VHH including in your table as well.
Author Response
Manuscript ID: antibodies-767324
Dear Editor,
Please find herewith our revised manuscript titled “Construction of antibody phage libraries and their application in veterinary immunovirology” authored by Shahbaz Bashir and Jan Paeshuyse. We have performed thorough revision according the suggestions of the Reviewers. Please find herewith a copy of the manuscript in which we highlighted the revisions. We responded as follows to the comments:
Reviewer#2:
- Abstract is not balanced:
While the body of the manuscript is well balanced now [between 1) the APD part and 2) APD application], the Abstract does not reflect that (the Abstract is by and large on the APD technology section). The authors should make the Abstract ~half and half. They should reduce the APD technology part and add for the Application part.
We now have modified the abstract ~half and half. The first half of the abstract deals with the antibody phage display and second half explains the veterinary medicine needs for phage display derived antibodies. This change is applied from line 17-23.
- Include this study if it fits
Nielsen K et al. Prototype single step lateral flow technology for detection of avian influenza virus and chicken antibody to avian influenza virus. J Immunoassay Immunochem. 2007;28(4):307-18.
We have added this reference. This study perfectly fits the scope of the review. This is a 36 reference number. (L214-216)
- “Construction of antibody phage libraries and its application in veterinary immunovirology”. Replace “its” with “their”
The word “its” has been replaced with “their” in the tile. Now the new title is
“Construction of antibody phage libraries and their application in veterinary immunovirology”
- P2, L42: Replace “similar” with “identical” in “ two similar light”
We have replaced “similar” with “identical”. (P1, L42)
- P2, L42: Replace “similar” with “identical” in “two similar heavy chains”
We have replaced “similar” with “identical” . Now the new sentence is
Their unique tetrameric structure consists of two identical light and two identical heavy chains, joined together by disulfide bridges and non-covalent interactions. (P1, L42)
- P2, L45: Delete “a” in “Antibodies are a modular protein defence system”
We have deleted “a”. Now the new sentence is: (P2, L45)
Antibodies are modular protein defence system
- Figure 1 legend: 1) Fc, is not fragment constant. It is fragment crystallizable;
We have corrected this into; Fc, fragment crystallizable (P2, L74)
2) VHH is not derived from the typical IgG as you imply. 3) Divide you figure into “a” and b”. For “a”, keep your conventional IgG, scFv, ScFv-Fc, Fab, F(ab)2 and scFab. Remove VH/VHH. For “b’, draw a typical camelid heavy chain IgG and from it show its VHH derivative.
We have divided the figure into A and B. “A”explains the conventional IgG with its formats such as scFv, ScFv-Fc, Fab, F(ab)2 and scFab. “B”explains the camelid HCAb separately with nanobody.
4) There are numerous disulfide linkages involved in these antibody fragments but you have chosen to show a few. I suggest you do not show any disulphide linkage and instead say: ”Inter- and intra-disulphide linkages are not shown”
We have removed all the disulfide linkages and wrote a sentence in the parenthesis “Inter- and intra-disulfide linkages are not shown here in the all formats”
; 5) “VHH, variable domain heavy chain of camels” this means it could also be derived from conventional camelid IgGs which is not true. Change to: “VHH, variable domain of camelid heavy chain IgGs”
We have drawn a new figure showing camelid heavy chain antibody alongside derived nanobody.
- p3, title: Add “display’ after “phage” in “Antibody phage library Construction and Biopanning”.
The word “display” has been added after “phage”. Now the title of heading is: (P2, L79)
“Antibody phage library Construction and Biopanning”.
- Antibody phage library Construction and Biopanning Section: the library construction description is very specific here: it is on ScFv and Fab, not on VH, VL or VHH libraries. It also describes immune and naïve libraries not synthetic libraries which does not involve the first few steps you are describing. It also described phage display libraries that use phagemid and not phage vector. These need to be acknowledged in the section to avoided misleading readers into understanding that what you are describing is general to all libraries.
We have added a sentence in L82 explaining that here are describing the antibody phage library construction of scFv and Fab format and not of VH,VL and VHH.
“Here, we shall briefly describe the construction of phagemid-based antibody (scFv and Fab) phage display library from immunized or naïve sources.”
We have also added a sentence regarding synthetic antibody phage libraries from L93-L95 with reference and avoided the confusion.
“In case of synthetic antibody phage libraries, initial few steps are not needed and library diversity is increased through the precise introduction of degenerate DNA into CDRs encoding region, thus rivalling or exceeding than that of natural immune repertoire”
- Figure 4:
It is a figure 2 now.
1) In the legend you have “PBMC” but in the figure itself “B-Lymphocytes”. Be consistent, stick to one.
In the figure, it has been added as PBMCs. So now in caption and figure both are consistent.
2) There is no background information on V/D/J and V/J and you do not need to include it. Remove any reference to V/D/J and V/J and stick only to VH and VL.
V/D/J and V/J fragments and annotation has been replaced with VH and VL.
3) In the right, Fab arm of your figure, remove “1” from “VL+CL1 gene pool”.
We have removed “1” from “VL+CL1 gene pool” and now it is “VL+CL gene pool”.
4) Replace “as” with “are” in “The amplified scFv and Fab as ligated”.
We have removed “as” with “are”. No it is “The amplified scFv and Fab as ligated”. (L102)
- Figure 2 legend. 1) Change “̴ 6.4 kb long” to “̴ 6.4 kb size”. 2) Include space in “DNA(ssDNA)”.
We have changed “̴ 6.4 kb long” to “̴ 6.4 kb size” also added space in “DNA(ssDNA)”.
Now the new sentence is:
It carries ̴ 6.4 kb size circular single stranded DNA (ssDNA).
It is not figure 2 anymore. It is figure 3 now.
- Figure 5 legend: 1) Add “of” after “generation” in “Panning of an antibody phage display library for the generation”.
It is figure 4 now. We have added “of” after “generation”. Now the new sentence is: (P5, L172)
Panning of an antibody phage display library for the generation of high affinity antibodies
- 2) Replace “The antigen is presented ... or expressed in cell surface” with “The antigen is presented ... or expressed on cells.”
We have corrected the sentence and now the new sentence is (L173-174)
The antigen is presented in its native form either immobilized on a plastic surface or expressed on cell surface.
- 3) Replace “Antibody library … through the process called washing.” With ““Antibody library … by washing.”
We have modified the sentence and now the new sentence is: (L175-176)
Antibody library displaying phages are incubated with antigen and non-bound or weakly bound phages are removed by washing
- 4) Replace “The tightly bound…by process called elution…” with “The tightly bound…are recovered by elution …”.
We have removed process called and now new sentence is: (P2, L176)
The tightly bound phages are recovered by elution
- 5) Replace “… 3-4 rounds leading to production of high affinity monoclonal antibody.” With “… 3-4 rounds leading to enrichment of binders.”
We have modified the sentence and now the new sentence is: (P2, L178-179)
This cycles continues for 3-4 rounds leading to enrichment of binders.
- Replace “The majors strides has been taken” with “The major strides have been taken”.
We have corrected the sentence and now new sentence is: (P6, L207)
“The major strides have been taken”
- Replace “Here, we mention the application of antibody” with “Here, we review the application of antibody”.
We have corrected the sentence and now new sentence is: (P6, L208)
Here, we review the application of
- Break “Here, we mention the application of antibody phage display derived antibodies in veterinary viral diagnosis, and therapeutics and immunoprophylaxis and these antibodies are also summarized in table 1 explaining the format, species of origin and target antigen.” in to sentences and modify: “Here, we review the application of antibody phage display derived antibodies in veterinary viral diagnosis, therapeutics and immunoprophylaxis. These antibodies are also described in table 1 with respect to format, species of origin and target antigen.”
We have corrected the sentence and now new sentence is: (P6, L208-210)
Here, we review the application of phage display derived antibodies in veterinary viral diagnosis, therapeutics and immunoprophylaxis. These antibodies are also described in table 1 with respect to format, species of origin and target antigen.
- Replace “murine single scFv capable of recognizing” with “ a murine scFv capable of recognizing”.
We have removed “single” and now new sentence is: (P6, L212)
murine scFv capable of recognizing
- Replace “Mining…has already yielded a novel scFv” with “Mining…yielded a novel scFv”
We have removed “has already” and now new sentence is: (P6, L217)
Mining of the rabbit antibody repertoire through antibody phage display yielded a novel scFv.
- Add “of” after “terminal” in “motif present at the C-terminal the F2 protein”
We have modified it and now the new sentence is: (P6, L218)
a conserved motif in the C-terminal of the F2 protein of Newcastle Disease Virus
- Modify “Besides murine and rabbit antibodies, chicken antibody repertoire itself has been explored with phage display for the screening of scFv against phosphoprotein (P) of NDV.” To “Chicken antibody repertoire has also been explored with phage display for the screening of scFv against phosphoprotein (NDV-P) of NDV.”
We have slightly modified it and now the new sentence is: (P6, L220-221)
Chicken antibody repertoire has also been explored with phage display for the screening of scFv against phosphoprotein (NDV-P) of NDV.
- Replace “Isolated scFv has shown higher specificity for NDV-P by ELISA and western blot [36].” With “An isolated scFv showed higher specificity for NDV-P by ELISA and western blot [36].”
We have slightly modified it and now the new sentence is: (P6, L222-223)
An isolated scFv showed higher specificity for NDV-P by ELISA and western blot
- This sentence is vague: “difficult to distinguish between very virulent IBDV (vvIBDV), classical, variant and vaccine strain…” Are all these four are different strains? If so, replace “strain” with “strains”.
Yes, these four are different strains of IBDV. (P6, L226)
We have replaced “strain” with “strains”.
- Break and modify “Although, new control programs have adopted a combined strategy of exclusion, slaughter and vaccination; however success rate depends upon accurate and reliable differentiation of vaccinated and infected animals.” To “New control programs have adopted a combined strategy of exclusion, slaughter and vaccination. However, success rate depends on accurate and reliable differentiation between vaccinated and infected animals.”
We have split the long sentence into two and now the new sentence is: (L239-L240)
New control programs have adopted a combined strategy of exclusion, slaughter and vaccination. However, success rate depends on accurate and reliable differentiation between vaccinated and infected animals.
- Replace “Previously through phage display, mice scFv was isolated” with “Previously through phage display, a murine scFv was isolated”.
We have replaced “mice” with “murine”. (P6, L242)
- Did you define FMDV before its mention in “against FMDV type O”?
We have now defined the FMDV (L237-238) with 7 serotypes: (P6, L239)
- Replace “In case of Bovine” with “In the case of Bovine”
We have added “the”in front of Bovine. (P6, L247)
- Replace “the initial pursuit of new diagnostic“ with “the initial pursuit for new diagnostic“.
We have replaced “of” with “for”. Now the new sentence is: (P7, L250)
In Pigs, the initial pursuit “for” new diagnostic tools using.
- Replace “Abs (rAbs) have been focused” with “Abs (rAbs) has focused”.
We have replaced “Abs (rAbs) have been focused” with “Abs (rAbs) has focused”. (P7, L250)
- “from camel immunized with commercial”. Is it camel or llama or alpaca? Also, in your table you only have camel as the source of all your VHHs. Is that true?
This specific anti- PVC2 antibody was obtained after immunization of bacterian camel.
While in the table 1, We have already changed it to the group camelids which covers camel, alpaca and llamas.
- “A single domain antibody (sdAb) was obtained from camel”. Use VHH here and in other places as they are all VHHs not other types of sdAbs. Also in your figure you specifically describe VHH only.
We have replaced “sdAb” with “”nanobody or VHH” in the whole text
- Replace “sdAb has already shown tremendous potential” with ““the VHH has already shown tremendous potential”. I am assuming here that you are referring to the VHH isolated above and not to sdAbs in general.
We have replace “sdAb” with the “nanobody”. We refer to the “VHH” and not the “sdAbs”in general. We have now clarified this confusion.
- “…as a promising experimental tool in imaging and porcine epidemic diarrhea virus (PEDV)”. Is “and” supposed to be “of”?
Yes, it should be “of”. We have corrected the sentence and added “of”. Now the new sentence is: (P7, L255)
“as a promising experimental tool in imaging of porcine epidemic diarrhea virus (PEDV)”
- “highly potent procinized murine antibody”. “procinized” or “porcinized”?
We have corrected it to be “porcinized”. Now it is: (P7, L256)
In addition to this, a highly potent porcinized murine antibody.
- Replace “swine influenza virus (SIV) and PEDV has” with “swine influenza virus (SIV) and PEDV have”.
We have replaced “has” with “have”. Now the sentence is: (P7, L258)
swine influenza virus (SIV) and PEDV have gained
- Replace “In Sheep” with “In sheep”.
We have replaced “In Sheep” with “In sheep”. (P7, L278)
- Replace “These scFvs have proved” with “These scFvs have been proven”.
We have replaced “These scFvs have proved” with “These scFvs have been proven”. (P78, L282)
- “…tool in control program of MVV.” “…tool in MVV control programs”?
We have modified the sentence and now it is: (P7, L283)
These scFvs have been proven to be an efficient research and diagnostic tool in MVV control programs
- Replace “veterinary therapeutics in very limited” with “veterinary therapeutics is very limited”.
We have replaced “veterinary therapeutics in very limited” with “veterinary therapeutics is very limited”. (P7, L287)
- Replace “It has shown promising therapeutic effects against” with “The ScFv has shown promising therapeutic effects against”.
We have replaced “It has shown promising therapeutic effects against” with “The ScFv has shown promising therapeutic effects against”. (P7, L295-296)
- “…specific nanobody was”. “VHH” instead of “nanobody”? See also other places for possible replacement of nanobody with VHH. If you are going to use “nanobody”, make sure you clarify it to be VHH in brackets in its first mention.
We have used the term “VHH” in all places and even in the first first mention we have explained it nanobody (VHH)
- Improve the structure of the sentence “The efficient binding alongside in vitro neutralization ability this anti-E2 nanobody make it a promising therapeutic candidate in controlling BVDV infection worldwide.”
We have improve the sentence and now the new sentence is: (P8, L307)
The efficient binding alongside in vitro neutralization ability make this anti-E2 nanobody a promising therapeutic candidate in controlling BVDV infection worldwide.
- Replace “respiratory and neurological disease” with “respiratory and neurological diseases”. There could be other places where you may need to replace “disease” with “diseases”
We have replaced “respiratory and neurological disease” with “respiratory and neurological diseases”.
We have also replaced disease replace “disease” with “diseases” in other places as well. (P8, L328)
- Replace “ways to control viral disease” with “ways to control viral diseases”
We have replaced “ways to control viral disease” with “ways to control viral diseases”. (P8, L330)
- Table 1. Be consistent: 1) Chicken and rabbit are singular, mice is plural. Replace “mice” with “mouse”.
We have replaced “mice” with “mouse”
- 2) If it is dog, then replace “canine” with “dog”;
We have replaced “canine” with “dog”.
- 3) since you use “diagnosis”, then replace “Therapeutics” with “Therapy”.
We have replaced “Therapeutics” with “Therapy”.
- 1) They cannot be all camel. At least some must be llama. Address all “camels”.
We have now used term “camelid” to rectify the confusion.
2) Replace “Synthetic human Griffin. 1“ with “human“, then in the footnote say: ”Synthetic human Griffin. 1” and include reference.
We have replaced “Synthetic human Griffin. 1“ with “human“ and also added the footnote explaining it with reference.
3) To be consistent with your antibody domain abbreviation style, e.g., VL, CL, CH1, use VHH instead of VHH including in your table as well.
To be consistent, we have “VHH” with “VHH” in the table 1.
We would like to thank the reviewers for their time and critical review of our manuscript.
Best regards,
Shahbaz Bashir, Jan Paeshuyse